

# Electroencephalographic modulations during an open- or closed-eyes motor task

Sébastien Rimbert[1,2,3], Rahaf Al-Chwa[1,2], Manuel Zaepffel[4] and Laurent Bougrain[1,2,3]

[1] Neurosys team, Inria, Villers-lès-Nancy, France
[2] Artificial Intelligence and Complex Systems, Université de Lorraine, LORIA, Vandœuvre-lès-Nancy, France
[3] Neurosys team, CNRS, LORIA, Vandœuvre-lès-Nancy, France
[4] Unaffiliated, Dambach-la-ville, France

## ABSTRACT

There is fundamental knowledge that during the resting state cerebral activity recorded by electroencephalography (EEG) is strongly modulated by the eyes-closed condition compared to the eyes-open condition, especially in the occipital lobe. However, little research has demonstrated the influence of the eyes-closed condition on the motor cortex, particularly during a self-paced movement. This prompted the question: How does the motor cortex activity change between the eyes-closed and eyes-open conditions? To answer this question, we recorded EEG signals from 15 voluntary healthy subjects who performed a simple motor task (i.e., a voluntary isometric flexion of the right-hand index) under two conditions: eyes-closed and eyes-open. Our results confirmed strong modulation in the mu rhythm (7–13 Hz) with a large event-related desynchronisation. However, no significant differences have been observed in the beta band (15–30 Hz). Furthermore, evidence suggests that the eyes-closed condition influences the behaviour of subjects. This study gives us greater insight into the motor cortex and could also be useful in the brain-computer interface (BCI) domain.

Corresponding author
Sébastien Rimbert,
sebastien.rimbert@inria.fr

## INTRODUCTION

Every day we accomplish voluntary movements in our environment, and yet the simplicity with which we perform these movements contrasts with the high complexity of their underlying physiological and neuronal processes. Voluntary movements, also called self-paced movements or self-initiated movements, can be subdivided into two categories: internal and external voluntary movements. For an internal voluntary movement, the motion is selected according to a defined goal, based on internal cognitive processes relative to our previous experience. This is contrary to a reflex response which is an immediate and stereotyped reaction to a stimulus from the environment (*Haggard et al., 2005*). For an external voluntary movement, the motor preparation phase is absent, but the phases of stimulus processing and anticipation are maintained (*Brass & Haggard, 2008*).

Various specific areas are associated with the motor control. For example, the pre-supplementary motor area is associated with selective motor initiation, and the medial pre-motor system is associated with internal timing (*Hoffstaedter et al., 2013*). In addition, a voluntary movement is part of a sensorimotor loop composed of a preparation, an execution, and an evaluation phase, each modulating the activity of different motor areas (*Deiber et al., 2012*). A self-paced movement is characterised by three distinct phases which are easily identifiable in the recorded EEG signal. Initially, compared to a resting state, the movement preparation phase shows a gradual decrease of power in the mu (7–13 Hz) and the beta (15–30 Hz) bands (*Pfurtscheller & Neuper, 1997*). This is referred to as an event-related desynchronisation (ERD) (*Pfurtscheller & Aranibar, 1979*; *Pfurtscheller & Lopes da Silva, 1999*). During the movement, a minimal power level is maintained in both bands (*Pfurtscheller & Lopes da Silva, 1999*; *Cheyne, 2013*). Finally, 300 to 500 ms after the end of the movement, there is an increase of power referred to as an event-related-synchronisation (ERS) in the beta band, also known as post-movement beta rebound, lasting approximately one second (*Salenius et al., 1997*; *Cheyne, 2013*; *Kilavik et al., 2013*). Concurrently, in the mu band, the power returns to a baseline after several seconds. Typically, these patterns occur in the motor cortex, in the contralateral position (*Pfurtscheller & Aranibar, 1977*; *Pfurtscheller & Neuper, 2001*; *Bai et al., 2005*) but can appear bilaterally (*Fok et al., 2011*; *Pfurtscheller, Stancák & Neuper, 1996*; *Formaggio et al., 2013*). ERD and ERS patterns are observed also during a passive movement (*Müller et al., 2003*), an observed movement (*Avanzini et al., 2012*), a kinesthetic illusion (*Keinrath et al., 2006*), a median nerve stimulation (*Salenius et al., 1997*) and a motor imagery (*Pfurtscheller & Solis-Escalante, 2009*). A self-paced movement affects not only the power in the mu and the beta frequency bands, but also the event-related potentials in the spatio-temporal domain (*Kornhuber & Deecke, 1965*; *Shibasaki & Hallett, 2006*).

Voluntary movements, such as grasping and manipulation of objects in our environment, require sensory inputs and feedback (*Desmurget et al., 1998*; *Jeannerod, 1995*). Indeed, directed movements towards an object are the result of transforming the visual properties of this object into a well-defined sequence of muscular contractions. Several studies (*Berger, 1929*; *Compston, 2010*; *Jasper, 1936*; *Barry et al., 2007*; *Barry et al., 2009*) show differences in resting state EEG signals recorded during eyes-closed (EC) and eyes-open (EO) conditions. These differences include an increase in alpha activity during an EC resting condition with respect to an EO condition with a visual stimulation (*Berger, 1929*). Moreover, the EC condition modulates the occipital, parietal and frontal alpha activities (*Li, 2010*) and the effect of the EO condition is most pronounced in posterior regions (*Chapman, Armington & Bragdon, 1962*; *Volavka, Matouek & Roubek, 1967*; *Legewie, Simonova & Creutzfeldt, 1969*).

Very few studies have investigated the influence of the EC condition on the motor cortex during a voluntary movement. One of these studies carried out by Westphal in 1993, found no difference between the EC and EO conditions compared to the *Bereitschaftpotential* (*Westphal et al., 1993*). However, to the best of our knowledge, no study has investigated the effect of the EC condition on the ERD and ERS patterns reflecting excitation and

inhibition occurring on the motor cortex. This raises the question of how the EEG power over the motor cortex are modulated before and after a voluntary movement during an EC condition.

In this work we analyse and compare the modulation of mu rhythm and beta band activities for a voluntary movement during EO and EC conditions. To this end, we recruited 15 healthy subjects and computed time-frequency maps, ERD and ERS modulation, and EEG power during resting state to show differences occurring on the motor cortex in the EEG signal. Our results indicate that the EC condition increased the ERD in the mu rhythm but had no effect on the ERD and ERS in the beta band.

## MATERIALS AND METHODS

### Participants

Fifteen right-handed healthy volunteer subjects (nine females; from 19 to 40 years-old; 24.1 years ±3.2) were recruited for this study. This experiment followed the statements of the WMA declaration of Helsinki on ethical principles for medical research involving human subjects (*World Medical Association, 2002*). In addition, participants signed an informed consent which was approved by the ethical committee of Inria (COERLE, approval number: 2016-011/01) as it satisfied the ethical rules and principles of the institute. The subjects had normal or corrected-to normal visual acuity and had no medical history which could have influenced the task, such as diabetes, antidepressant treatment or neurological disorders.

### Experimental task

The task was an isometric flexion of the right index finger in order to click on a computer mouse during two different conditions: EC and EO. Although simple to execute, this kind of finger movement has been found already to activate the motor cortex (*Shibasaki et al., 1993*). To ensure that this movement was a voluntary one, the protocol dictated that the subject performed the task at their own pace, without receiving a triggering signal.

### Experimental environment

The experiments took place in a confined and calm room and were conducted by the same researcher. Subjects were seated comfortably on a chair with their right arm at their side with their hand placed on the computer mouse on a table without any tension of the muscles.

### Experimental design

The protocol contained two conditions (EC and EO) split in different runs that were completed by the subject on the same day. Condition 1 and condition 2 corresponded to the voluntary movement in EC and EO conditions respectively (Fig. 1). For each condition, 3 runs of 5 min were performed and a randomisation of the runs was applied to avoid that fatigue, gel drying, or other confounding factors that might have caused possible biases in the results. During a run, the subject had to perform a real movement task approximately once every ten seconds. At the beginning of each run, the subject remained relaxed for 10 s. Breaks of a few minutes were taken between runs to prevent fatigue of the subject.

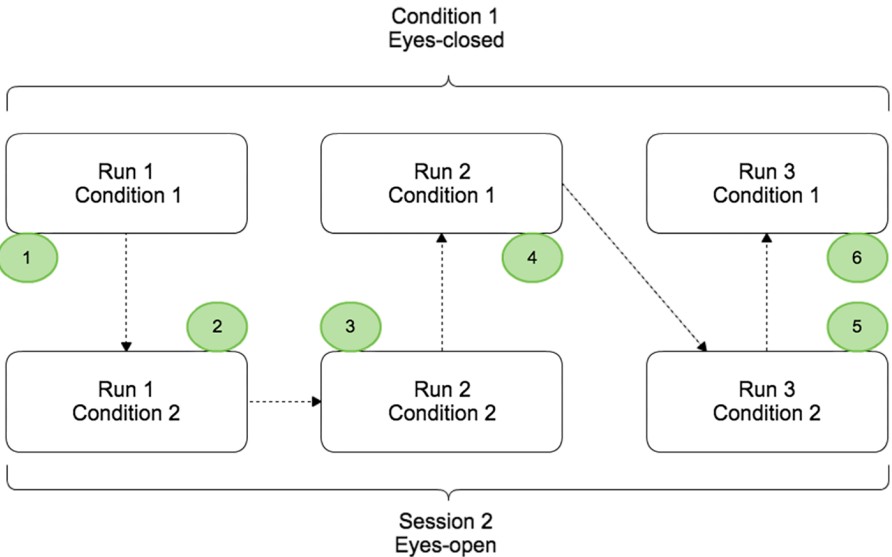

**Figure 1  Paradigm scheme representing the two different conditions: EC (Condition 1) and EO (Condition 2).** Each condition was composed by three runs of 5 min and randomisation of the order was applied.

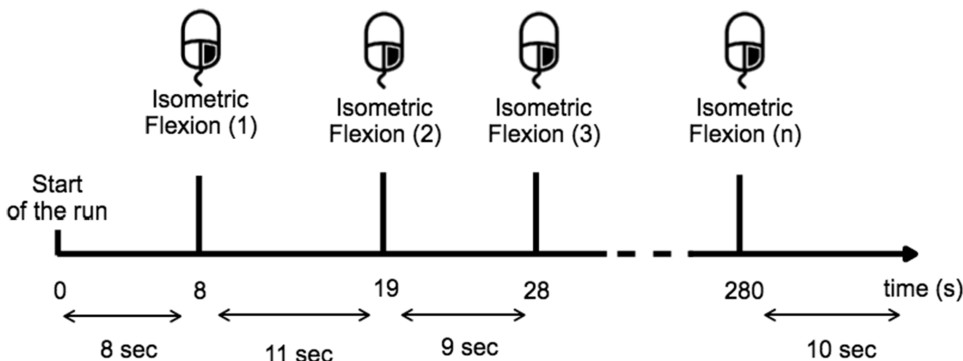

**Figure 2  Timing scheme for each run.** The subject performed a right-hand index isometric flexion on the mouse when they wished while providing sufficient time (>8 s) between each movement.

Before the experiment, the task was described and the subject was trained to execute it over 10 min. During the training period the subject learned to estimate 10 s in the two given conditions and for the open-eyes condition the subject had to perform the task while they fixate their gaze on a target placed at eye-level to avoid random eye movements (Fig. 2). The subjects were instructed to avoid swallowing or any other movements while performing the task, particularly all eye-movement.

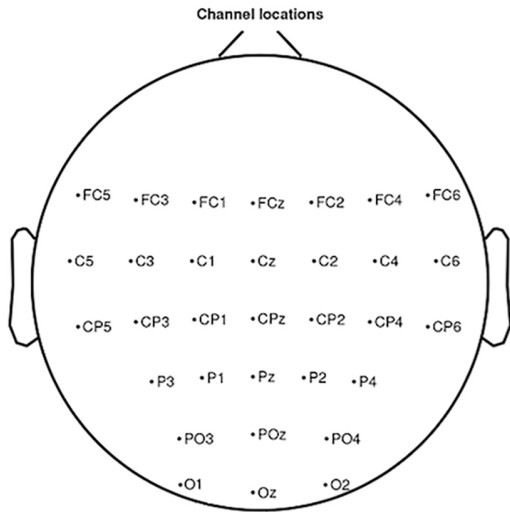

**Channel locations**

**Figure 3** Electrode positions localised around the primary motor cortex, the motor cortex, the somatosensory cortex and the occipital cortex in accordance with the international 10–20 system.

## Physiological recordings

A custom-written scenario for the open source software OpenViBE (*Renard et al., 2010*) was designed to automate the recording of EEG signals and was triggered during the task. EEG signals were recorded with a BiosemiTM Active Two 32-channel EEG system at 2,048 Hz.

In the Biosemi$^{TM}$ system the ground electrodes used were two separate electrodes: Common Mode Sense (CMS) active electrode and Driven Right Leg (DRL) passive electrode. The EEG was recorded from 32 sites in accordance with the international 10–20 system: $FC_5$, $FC_3$, $FC_1$, $FC_z$, $FC_2$, $FC_4$, $FC_6$, $C_5$, $C_3$, $C_1$, $C_z$, $C_2$, $C_4$, $C_6$, $CP_5$, $CP_3$, $CP_1$, $CP_z$, $CP_2$, $CP_4$, $CP_6$, $P_3$, $P_1$, $P_z$, $P_2$, $P_4$, $PO_3$, $PO_z$, $PO_4$, $O_1$, $O_z$, $O_2$ (Fig. 3). These sites are localised around the primary motor cortex, the motor cortex, the somatosensory cortex and the occipital cortex, which allowed us to observe the physiological changes due to the task for the open eyes condition. The setup of the EEG channels was the same in both EC and EO conditions. An external electromyogram (EMG) electrode was added in order to measure the extensor indicis muscle activity. For some subjects the measured EMG activity was less reliable than the trigger obtained by mouse click, which is why we used this trigger for the analysis. Impedance was kept below 10k $\Omega$ for all electrodes to ensure that the background noise in the acquired signal was low. No additional filtering was used during the recording.

## Signal pre-processing

All offline analyses were performed using the EEGLAB toolbox (*Delorme & Makeig, 2004*) and Matlab2016a (The MathWorks Inc. Natick, MA, USA). The data were processed in General Data Format (GDF). First, raw EEG data were filtered, transformed, and Laplacian processing was applied to minimise the noisy EEG signals (*Perrin et al., 1989*). Then, EEG

signals were resampled at 256 Hz and divided into 6 s epochs corresponding to 2 s before and 4 s after the mouse click for each run. Trials contaminated by excessive noise or movement artefacts were removed. Baseline was defined 2 s before each trial, therefore a specific baseline was chosen for the two conditions. Finally all runs were merged for a given condition for each subject. In this article, we chose the $C_3$ electrode because it corresponds to the motor area of the right hand control.

These electrodes are commonly analyzed in motor control studies since they are located above cortical regions known to be involved in movement preparation and execution

## Spectrum

The power spectrum was computed with EEGLAB (Fig. 4A) and showed the signal power distribution along a range of frequencies spanning (8–35 Hz).

## Time-frequency analysis

To analyse the differences between both conditions, we performed an event-related spectral perturbation (ERSP) analysis between 8–35 Hz with a resolution of 0.5 Hz at 13.68 ms intervals. We used a 256 point sliding fast Fourier transform (FFT) window with a padratio of 4 and we computed the mean ERSP 1.5 s before the task to 3.5 s after the task. ERSP allows to visualise event-related changes in the average power spectrum relative to a baseline (2 s) interval taken 2 s before each trial (*Brunner, Delorme & Makeig, 2013*). Time-frequency analysis is very useful to know precisely which frequency band will be used to analyse the results. As we can see in Fig. 4B, mu rhythm (10–13 Hz) and beta band (15–30 Hz) are modulated strongly, therefore, we used these frequency bands for plots and other figures such as the time frequency representation or the topographic maps of the brain.

## Topographies

Brain topography allowed us to display the possible changes over different electrodes on the scalp in order to localise which part of the brain was involved when the subject performed the requested task. Two bands of frequencies were used to compute ERSPs: alpha/mu (10–13 Hz, Fig. 5A) and beta (15–30 Hz, Fig. 5B) for EC and EO conditions.

## ERD/ERS quantification

We computed the ERD/ERS% using the "band power method" (*Pfurtscheller & Lopes da Silva, 1999*).

$$\mathrm{ERD/ERS\%} = \frac{\overline{x^2} - \overline{BL^2}}{\overline{BL^2}} \times 100, \tag{1}$$

where $\overline{x^2}$ is the average of the squared signal smoothed using a 250-millisecond sliding window with a 100 ms shifting step, $\overline{BL^2}$ is the mean of a baseline segment taken at the beginning of the corresponding trial, and ERD/ERS% is the percentage of the oscillatory power estimated for each step of the sliding window. A positive ERD/ERS% indicates a synchronisation whereas a negative ERD/ERS% indicates a desynchronisation. This percentage was computed separately for all EEG channels. The EEG signal was filtered in the mu rhythm (10–13 Hz) and in the beta band (15–30 Hz) for all subjects using a 4th-order Butterworth band-pass filter.

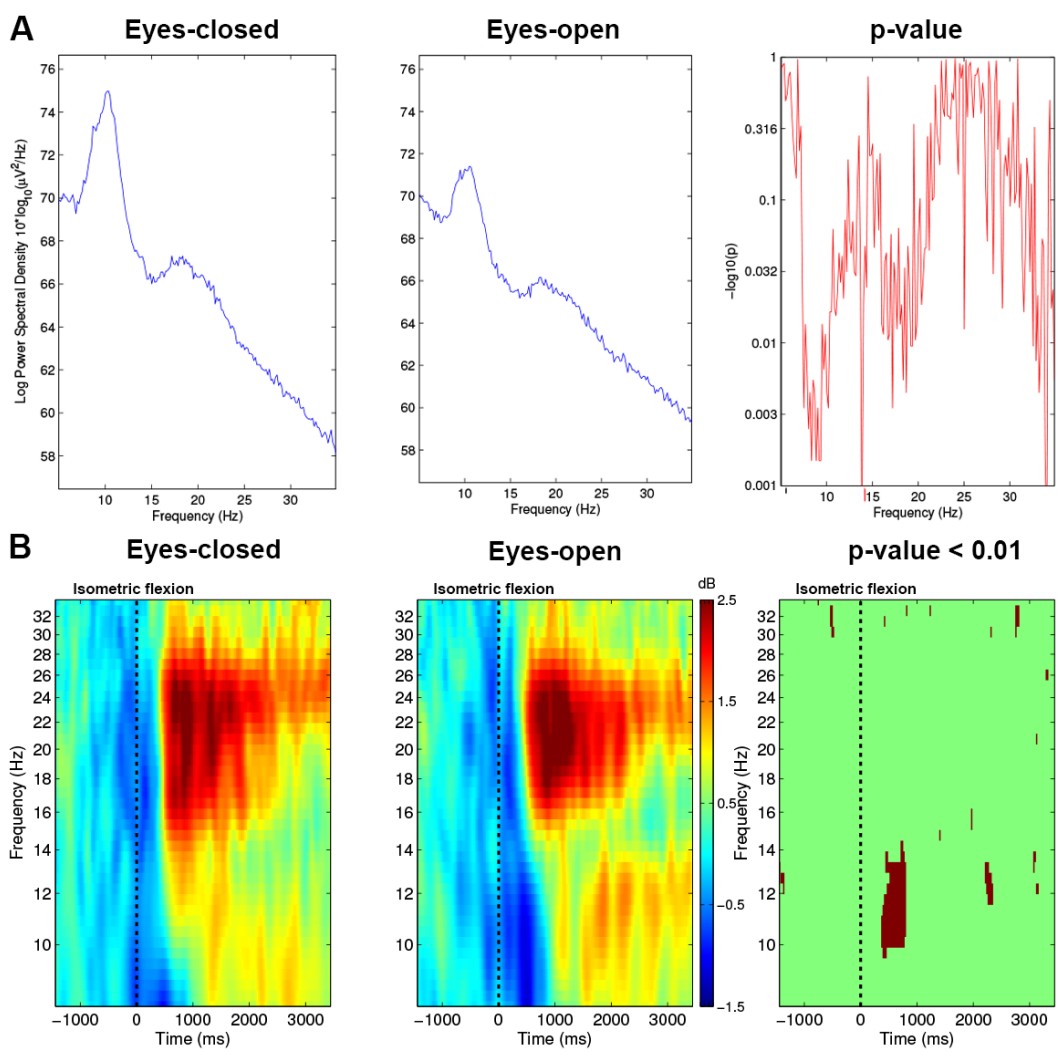

**Figure 4** (A) Spectrum analysis grand average ($n = 15$) difference between Condition 1 (EC) and Condition 2 (EO) for electrode $C_3$. (B) Time-frequency grand average analysis (ERSP) for Condition 1 (EC) and for Condition 2 (EO) for electrode $C_3$. A red colour corresponds to strong modulations in the band of interest. Significant difference ($p < 0.01$) are shown in the final part of the figure.

ERD and ERS are difficult to observe from the raw EEG signal. Indeed, an EEG signal expresses the combination of activities from many neuronal sources. We used the averaging technique to represent the modulation of power of the mu and beta rhythms during both conditions (Fig. 6) since it is considered one of the most effective and accurate techniques used to extract events (*Quiroga & Garcia, 2003*; *Pfurtscheller, 2003*).

## Statistical analysis

We chose to apply a paired $t$-test (two-sided) to show the significant difference about number of clicks performed shown on Fig. 7 ($p$-value $< 0.01$). The same $t$-test was applied on three chosen parts of the EEG signal, pre-movement phase (pre-M) ($-2,000$ ms; 0 ms), movement phase (M) (0 ms; 500 ms), and post-movement phase (post-M) (500 ms;

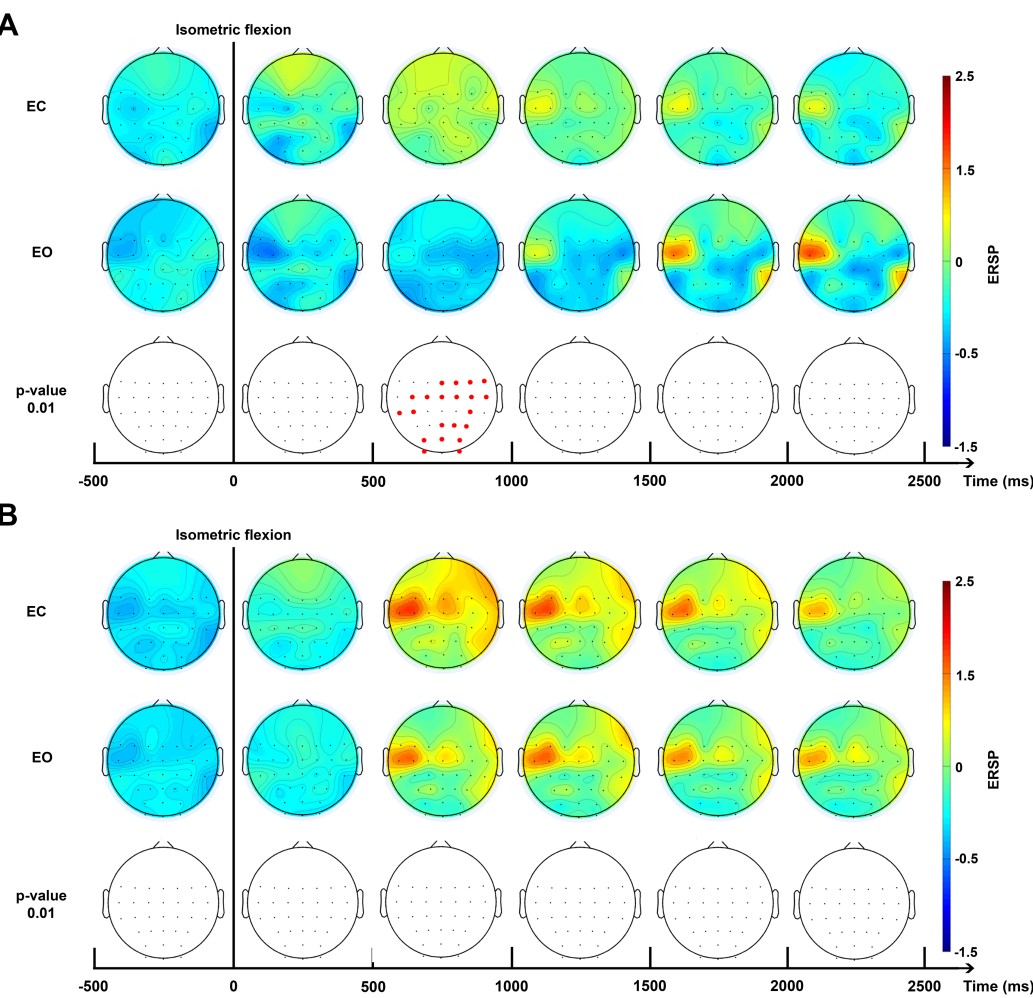

**Figure 5** **Topographic map of ERD/ERS% (grand average, $n = 15$) in the Alpha/mu band (A, 10–13 Hz) and in the beta band (B, 15–30 Hz) during two conditions: eyes-closed (EC) and eyes-open (EO).** A red colour corresponds to a strong ERS and a blue one to a strong ERD. A black line indicates when the isometric flexion started. This figure is an extrapolation through 32 electrodes. Red electrodes indicate a significant difference ($p < 0.01$).

2,000 ms) shown on Fig. 8 (*p*-value $< 0.01$). These parts were selected specifically by using the literature and our results for EO and EC conditions (*Pfurtscheller & Solis-Escalante, 2009*; *Pfurtscheller, 2003*; *Avanzini et al., 2012*; *Kilavik et al., 2013*).

A surrogate permutation test ($p < 0.01$; 2,000 permutations) from the EEGLAB toolbox was used to validate differences in terms of time-frequency and localisation of this ERSPs (Figs. 4 and 5). In addition to this treatment, we applied a false discovery rate (FDR) correction test in order to clarify how the false discovery rate was controlled for multiple comparisons. This test consisted of repetitively shuffling values between conditions and recomputing the measure of interest using the shuffled data. This test performed the drawing of data samples without replacement and is thought to be appropriate to show the difference between EC and EO conditions (*Manly, 2006*).

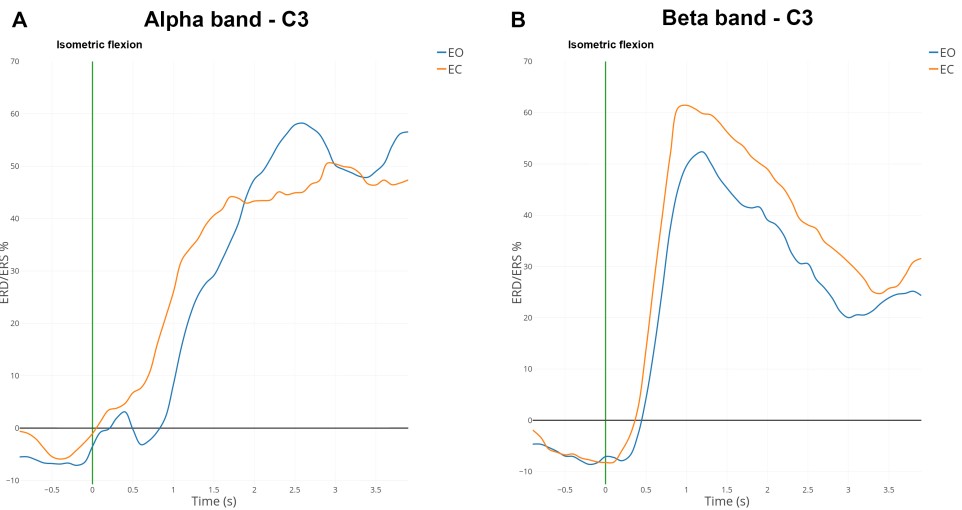

**Figure 6** Grand average ($n = 15$) ERD/ERS% curves in the mu (A) and the beta (B) bands for EO (in blue) and EC (in orange) conditions for electrode $C_3$. The green bar at 0 s corresponds to the voluntary movement performed.

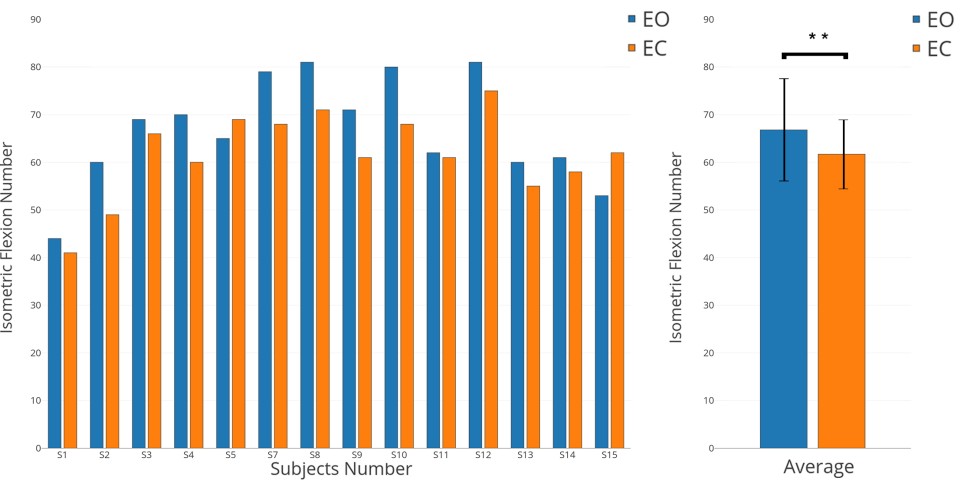

**Figure 7** Individual and average results of number of voluntary movements in EO (in blue) and EC (in orange) conditions.

## RESULTS

### Spectrum and time frequency

With the EC condition, significantly higher power values (**, $p < 0.01$) of the spectrum were measured on $C_3$ for a real movement in the upper mu rhythm (10–13 Hz) and low beta band (15–25 Hz) (Fig. 4A).

In Fig. 4B, before the movement, a decrease of ERSP (in blue) appeared in the mu rhythm, which started 400 ms before the execution of the motor task. From 0 to 500 ms, this desynchronisation remained present in the mu and beta frequency bands but was

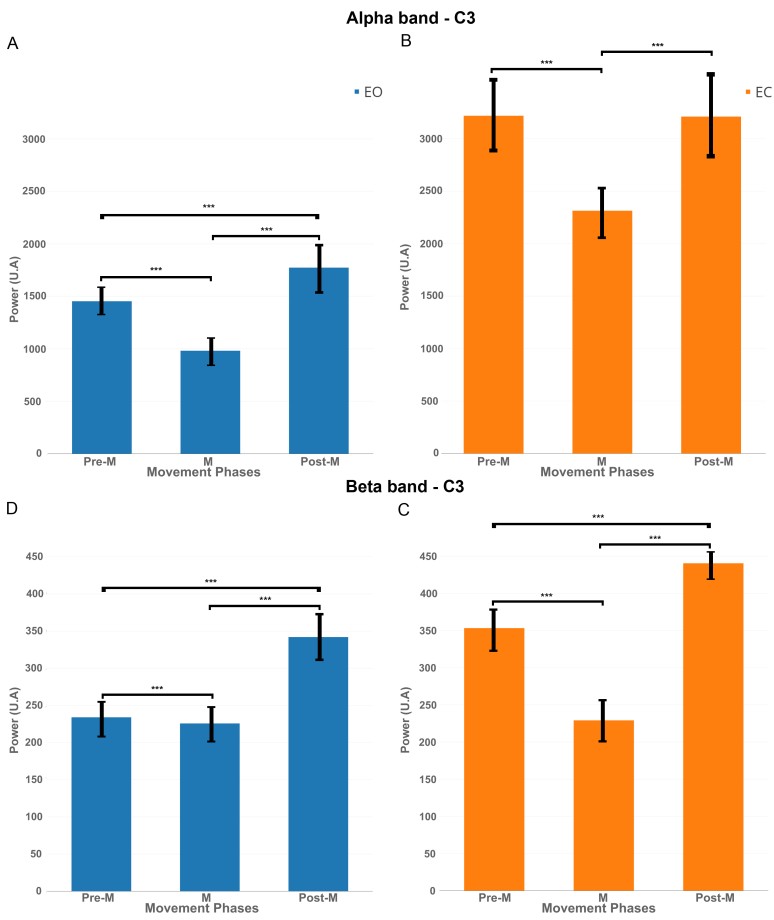

**Figure 8** EEG signal power (grand average, $n = 15$) in the alpha band (A and B, 7–13 Hz) and in the beta band (C and D, 15–30 Hz) for both conditions: EC (in orange) and EO (in blue) for the electrode $C_3$ in the three phases of voluntary movement: pre-movement (pre-M), movement (M) and post-movement (post-M). *** corresponds to a $p$-value $e < 0.001$.

stronger in the mu rhythm. In this time interval, Fig. 4B shows on the right side a significant difference with a $p$-value $< 0.01$ for the EO condition vs the EC condition comparison on the electrode $C_3$.

In the beta frequency band, at 1,000 ms, both conditions showed a similar increase (in red) but in the EO condition a residual rebound is maintained in the mu frequency bands. This rebound was absent in the EC condition. These differences seem not to be significant at $p$-value $< 0.01$ (Fig. 4B; right-side).

## Topography

The time-frequency analysis of ERSPs (Fig. 4B) showed powerful differences in the upper mu band (10–13 Hz) between EO and EC conditions. This allowed us to select two specific frequency bands for the topographic figures.

For the higher mu band (10–13 Hz), before the movement, there was no detectable difference between the EC and EO condition (Fig. 5A). Interestingly, 500 ms after the

voluntary movement, a significant difference of ERSP was detected on several bilateral electrodes, specially close to the motor cortex and the somatosensory motor cortex. Moreover, a stronger desynchronisation can be observed in the EO condition located in the motor cortex on the contralateral side with these changes also appearing in the occipital area. Conversely, a synchronisation was observed in the EC condition located on the somatosensory contralateral cortex. At 1,000 ms, a transient generalised synchronisation of the motor cortex was observed bilaterally and centrally for the closed eyes condition. At this time, for the EO condition, the synchronisation was located only on the contralateral side of the motor cortex; therefore, it was significantly different from the EC condition on the central and ipsilateral side. From 1,500 ms to 4,000 ms no significant changes were observed, but the synchronisation appeared to be stronger in the EO condition on $C_3$ which represents, contralaterally, the area of the hand on the motor cortex.

No significant difference was observed in the beta frequency band (15–30 Hz) between both conditions (Fig. 5B) and confirmed results showed in Fig. 4B. A contralateral synchronisation first occurred and it spread out over the central area near by the $C_3$ and $C_5$ electrodes. This synchronisation started around 500 ms and maintained itself until 1,500 ms and then slowly disappeared while still remaining very visible at 4,000 ms.

## ERD and ERS modulations

ERD and ERS modulations were computed in the mu band (7–13 Hz) and the beta band (15–30 Hz) for all subjects. Figure 6 shows the grand average of ERD/ERS% curves over 15 subjects on electrode $C_3$.

In the mu band, the desynchronisation described in the literature appeared one second before the voluntary movement (i.e., isometric flexion). An interesting synchronisation started at 500 ms to reach a maximum of 50% for the EC condition or 60% for the EO condition (Fig. 6A). A total of 500 ms after the self-paced movement in EO condition, there was a desynchronisation which was absent for EC condition. This difference was observed already on the spectrogram (Fig. 4B) and topographic map (Fig. 5A).

In the beta band, there was a desynchronisation which appeared one second before the voluntary movement and remained present 500 ms afterwards (Fig. 6B). One second after the voluntary movement, the power in the beta band increased by around 60%, reached its maximum and returned to the baseline after 4 s. The evolution from ERD to ERS was rapid (less than one second) and should be linked to the type of movement (isometric flexion of right index). For the EC condition the synchronisation seems to be slightly higher than the opposite condition but is not statistically significant.

## The number of movements performed decreases with eyes-closed

While all subjects received the same instructions, were trained for both conditions (EO and EC), and performed all runs in randomised order, a significant behavioural difference was observed. Indeed, the numbers of isometric flexions performed in both conditions were different (Fig. 7, $p$-value $< 0.01$). With some exceptions (S5 and S15), all subjects performed more voluntary movements when they had to keep their eyes open. In average, 66.8 clicks were executed for the EO condition versus 61.7 for the EC condition.

## Variations of EEG power during pre-movement, movement and post-movement phases

In order to better understand the influence of the EC condition on the motor cortex, we computed EEG power in the mu band (7–13 Hz) and beta band (15–30 Hz) for the electrode $C_3$ during three phases: Pre-movement (−2,000; 0 ms), Movement (0; 500 ms) and Post-Movement (500; 2,000 ms).

Figure 8A shows that, for the EO condition, there was a significant decrease of power ($p < 0.01$) in the mu band between pre-M and M phases. There was an increase of power between M and post-M phases which was also significant compared to pre-M phase.

Figure 8B shows for the EC condition that there was also a significant decrease of power between pre-M and M phases but the power returned back to the pre-M phase level and did not increase significantly under this condition.

The same pattern was reflected significantly ($p$-value $< 0.01$) in Figs. 8C–8D in the beta band, namely a decrease between pre-M and M phases and an increase in post-M phase but here, the decrease of power was more obvious for the EC condition.

Finally, it is interesting to note that the power was higher for the EC condition in both frequency bands, so the decrease of power was more obvious in this condition.

## DISCUSSION

### Differences observed between eyes-closed and eyes-open condition during the voluntary movement

In this study, the differences observed between EC and EO conditions for a voluntary movement task prompted the question of the sources of these differences. Two possibly complementary hypotheses explain the differences obtained in terms of ERD and ERS (Fig. 4B) and behaviour (Fig. 7):

- The EC condition creates a strong modulation of the mu rhythm (7–13 Hz) and disturbs the EEG signal over the whole cortex;
- The EC condition involves a behavioral change during a voluntary movement and modulates the activation/deactivation of the motor cortex.

The first hypothesis suggests that results described in this study were a consequence of global disruption generated by EC condition in the EEG signal. Moreover, the major difference observed was the large ERD in the mu rhythm for the EO condition (Fig. 4B). Several studies (*Barry et al., 2007*; *Barry et al., 2009*; *Westphal et al., 1993*; *Legewie, Simonova & Creutzfeldt, 1969*) have described differences occurring only in the alpha band and support this hypothesis.

The second hypothesis can be complementary to the first hypothesis, suggests that when a voluntary movement is performed, the preparation, the execution and the feedback phases linked to this movement can change according to EC and EO conditions. Our results showed that the number of voluntary movements performed is different for both conditions (Fig. 7). This shows that the internal representation of time is not similar between EC and EO conditions. This has been confirmed also by several studies showing that opening and closing eyes are fundamentally different behaviours (*Marx et al., 2004*;

*Liang et al., 2014*). More specifically, there is an "interoceptive" mental activity identified by imagination and multisensory activity during the EC condition (*Miraglia et al., 2016*; *Marx et al., 2004*). The post-testimonies of subjects support this idea. Therefore, in EC condition, preparation and feedback phases could be different and that will have consequences on the activation/deactivation of the motor cortex. In the literature, modulations of both mu and beta bands are involved in a motor task (*Kilavik et al., 2013*; *Pfurtscheller & Lopes da Silva, 1999*; *Avanzini et al., 2012*; *Neuper & Pfurtscheller, 2001*). Although only changes in the mu band were observed (Fig. 4), and support the first hypothesis, other results (Fig. 8) showed clearly differences in term of EEG power for movement phases (Pre-M, M and Post-M) between both conditions.

These results suggest that the mechanism on the motor cortex is different between the two conditions, particularly in EC condition because there was no synchronisation in the mu rhythm but a synchronisation in the beta band (Fig. 8). This hypothesis is supported by a recent paper published by *Cambieri et al. (2017)* showing that the excitability of the motor cortex is the same for EC and EO.

## ERD and ERS modulations for EO and EC conditions

The results are coherent with previous studies describing ERD/ERS% modulations in the mu and the beta bands during motor actions (*Pfurtscheller & Aranibar, 1979*; *Kilavik et al., 2013*; *Avanzini et al., 2012*; *Shibasaki et al., 1993*). The low power of the ERD (Fig. 6) can be explained because subjects were instructed to focused more on the precision rather than the speed of the movement (*Pastötter, Berchtold & Bäuml, 2012*). However, although subjects were consciously making an effort to carry out a voluntary movement, we must consider that clicking on a mouse could be somewhat of a reflex due to habit. This could have an impact on the low ERD amplitude.

The desynchronisation in the mu rhythm (Fig. 6) started 2 s before the voluntary movement and was bilateral and not predominant on the ipsilateral position (*Pfurtscheller & Aranibar, 1979*). This result suggests a bilateral activation of the motor cortex during an unilateral movement (*Formaggio et al., 2013*). Interestingly, the stronger desynchronization was found between 10–13 Hz and corresponds to the mu rhythm (*Pfurtscheller, 2001*). This shows that this results are due from the combination of a change in the alpha band (over the occipital region) under to EC condition and a change in the mu (over the motor cortex) due to the movement performed in EC condition.

No major differences were observed in the beta band between EO and EC conditions. This result confirms that the beta band is not modulated, in term of ERDs and ERSs, depending on the experimental conditions (*Kilavik et al., 2013*).

## Role in the BCI domain

The results obtained could be helpful for the Brain–Computer Interface (BCI) domain. Next step in this direction would be to verify that modulations observed for the EC condition are confirmed during an attempted movement or a motor imagery.

Similarly, this finding could be used for motor rehabilitation after a stroke with a BCI (*Cincotti et al., 2012*). These specific BCIs normally use visual feedback to inform the user

about system's decisions after a motor imagery task. However, performing the motor task with closed-eyes with a tactile (*Jeunet et al., 2015*) or an auditory feedback (*Nijboer et al., 2008*) could be considered to improve concentration and attention during a motor task (*Egeth & Yantis, 1997*; *Kim & Cruz, 2011*). In addition, the EC condition would solve the problems of gaze fixation and artifacts caused by the EO condition. In order to progress towards this hypothesis, it would be necessary to carry out further experiments and establish a link between learning, cerebral plasticity and the EC condition.

## CONCLUSION

In this article, we compared modulations in the EEG signal over the motor cortex correlated with a voluntary movement for two conditions: eyes-closed and eyes-open. We showed that a greater desynchronisation appeared 500 ms during the voluntary movement in the mu rhythm (10–13 Hz) for the EC condition. We found also that there was no significant difference in the beta band (15–30 Hz). Furthermore, the number of voluntary movements was significantly different between the two conditions and showed that the closed eye condition influenced behaviour of the subjects. This study gives us greater insight into the motor cortex and could also be useful in the BCI domain.

### Funding
The authors received no funding for this work.

### Competing Interests
The authors declare there are no competing interests.

### Author Contributions
- Sébastien Rimbert and Rahaf Al-Chwa conceived and designed the experiments, performed the experiments, analyzed the data, contributed reagents/materials/analysis tools, prepared figures and/or tables, authored or reviewed drafts of the paper, approved the final draft.
- Manuel Zaepffel and Laurent Bougrain analyzed the data, contributed reagents/materials/analysis tools, prepared figures and/or tables, authored or reviewed drafts of the paper, approved the final draft.

### Human Ethics
The following information was supplied relating to ethical approvals (i.e., approving body and any reference numbers):
The ethical committee of Inria (COERLE) granted approval to carry out the study within its facilities (approval number: 2016-011/01).

### Data Availability
The raw data for Figs. 7 and 8 have been provided as Supplemental Information 1.

## Supplemental Information

Supplemental information for this article can be found online at http://dx.doi.org/10.7717/peerj.4492#supplemental-information.

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
