# Peer review of "Electroencephalographic modulations during an open- or closed-eyes motor task"

_PeerJ, doi:10.7717/peerj.4492_

## Round 0.1 · original submission · Major Revisions

· Academic Editor

Major Revisions

The authors should clarify how the false discovery rate was controlled for multiple comparisons. I recommend adding scatter plots to all bar graphs to visualise individual data points (see e.g. https://doi.org/10.1371/journal.pbio.1002128). Publication in PeerJ does not depend on novelty or impact. However, the presented results should be appropriately interpreted in relation to existing literature. It is well established that alpha oscillation increase in eyes-closed condition and it should be clarified how this can be used for BCI applications. With regards to data sharing, it should be noted that several public repositories such as Zenodo or OSF allow large data to be freely uploaded. Please ensure that the English language in the manuscript meets our standards, as PeerJ does not offer copyediting.

Reviewer 1 ·

Basic reporting

The paper would benefit from English language editing.
Please also include relevant references to other neural correlates of self-paced movement than the ERD/S that are represented over the motor cortex.
The figures have a good quality; however, all the figures are in the methods section. I would suggest to restructure the methods and the results sections.

Experimental design

no comment

Validity of the findings

no comment

Annotated reviews are not available for download in order to protect the identity of reviewers who chose to remain anonymous.

Reviewer 2 ·

Basic reporting

The manuscript discusses the impact of closing or opening one’s eyes when performing a finger movement task on the oscillatory components of the EEG signals. The well-written manuscript carefully explains the central research question, methods used and results. A sufficient literature background is also provided and the manuscript follows the professional article structure. However, the authors should address several points to improve the manuscript.

I would like to remark that the raw EEG data should be shared, too. The authors write that the EEG data is too large to be made available. This is hard to believe for me with the small number of subjects, short sessions and a moderate number of EEG channels and an invalid reason according to the PeerJ guidelines.

Experimental design

The research question is relevant, well-defined and meaningful. The methods were mostly well-explained. Concerning Figure. 5, a paired t-test was applied to detect significant differences in the ERD/ERS% changes for EO and EC. Regarding the validity of the tests, I believe that the repeated testing procedure leads to an unrealistic high number of significant differences. A correction for multiple testing should be applied here.

I also cannot match the results from Figure 5 and 6. In Figure 5, the ERD/ERS% values go up to 2.5 and 1.8, respectively whereas they go to 60 in Figure 6. Can you please explain the differences in the y scale? I would also be nice to have the same color scale on both subplots in Figure 5.

Two more small comments:

line 122: It is unclear to me what you mean by “Laplacian processing”.
line 186: It is rather unusual to use three asterisks *** for p<0.01. Normally, one uses (* for p<0.05, ** for p<0.01, *** for p<0.001).

Validity of the findings

The conclusion of the work is sound and linked to the original research question.

As an addition to all the grand average result, I would like to see single-subject results. How many of the subjects showed a significant difference in the 10-13Hz band between EO and EC? Sometimes grand average results might be misleading when they are driven by individual subjects with very strong signals.

Additional comments

The authors discuss the meaning of motor imagery for BCIs used for assessing intraoperative awareness and motor rehabilitation after stroke. I believe that the authors can make an important contribution to the BCI community by comparing the information content of the motor imagery events for the EO and EC condition. Using machine learning methods from BCI, I would like to see a comparison of how well an MI event can be discriminated from a random time segment for both conditions. If there are any significant differences between the two conditions then the authors can comment on which condition gives better control commands in a BCI application. The stronger ERD in the alpha band of the EC condition makes me believe that the signals are more informative in that case.

Minor comments:

The abstract should focus more on the differences between the ERD/ERS changes in the EO and EC condition.
line 62-63: Grammar error: The structure of the sentence in brackets is wrong.
line 85: The use of the word “session” is slightly misleading. If I understood the experimental design correctly, then there was no (larger) break between the sessions/runs. Normally, the word session is used for whole recording units and sessions are separated by larger breaks.
line 93: Grammar error: “were takes”
line 107: Explain the abbreviations “CMS” and “DRL”.
line 216: You wrote OE instead of EO.
Caption of Figure 6: The authors should abbreviate both EO and EC or none of them.
line 270: “figure” should be written as “Figure”
line 282: Typo: “theses”.
line 296: A verb is missing in the sentence.
References 26-27: “EEG” should be written in capital letters

·

Basic reporting

The article is generally well written although I recommend a further check with a native English (eg. page 2, beginning of the intro: Every day (I would omit comma); page 2, line 29 can appearS bilaterally..).

References to relevant literature investigating cortical excitability under EC condition is missing (eg. Boroojerdi ed al. 2000, Fierro et al., 2005, Cambieri et al. 2017) and could potentially improve the interpretation of the results.

The opening statement in the abstract "many studies have shown that EEG is modulated by the EC condition..." left me a little unsure: it is not only studies that have shown so, but rather it is a basic knowledge in neurophysiology that EEG reacts to EC/EO (alpha rhythm reactivity is probably the first notion any neurophysiologist learn).

Experimental design

The experimental design is clear and well described.

I am not sure that the performed analysis on EEG (erd/ers) is the most suitable instrument to evaluate differences in the two conditions, for the same reason mentioned above: a spectral analysis performed in the alpha and beta ranges (erd/ers) is greatly influenced by a MAJOR event which is known to occur in the EEG (alpha rhythm reactivity).

Validity of the findings

Regarding the findings, my concerns are mainly those expressed above: differences between EC/EO conditions were found only in alpha band (which is a major, visible event occurring in the EEG); I cannot prevent myself from questioning the novelty or interest of the finding.

I do question whether this type of analysis/instrument is suitable to investigate changes in the motor cortex in the two conditions (since others have shown that there is an influence eg. Boroojerdi ed al. 2000, Fierro et al., 2005, Cambieri et al. 2017).

I am also unsure of the speculative role proposed in the BCI domain: the first proposed setting is a BCI monitoring during general anesthesia, which is a VERY different condition with respect to an awake person with closed eyes (so many other factors other than EC/EO...). The second proposed setting is a BCI for motor rehabilitation, in which I do question the added value (concentration?) offered by an EC paradigm (an approach which is to my knowledge very far from the ecological approach to -motor- rehabilitation).

Additional comments

The article is clear and well written. I do question the interest and novelty of the findings, which are possibly limited by the chosen analysis method. I also fail to see the possible utility of the results in the proposed BCI settings.
I strongly suggest to the authors to conclude the article with an anticipation of their next steps (in terms of: further analyses and/or examples of application in the short term).

---

## Round 0.2 · Minor Revisions

· Academic Editor

Minor Revisions

The reviewers were largely satisfied with the revised manuscript, but identified a few outstanding issues that need to be addressed before the manuscript can be accepted.

Reviewer 1 ·

Basic reporting

Although, the authors addressed my concerns satisfactorily, I still have some suggestions for further improvement.

I believe it would be beneficial also for the future readers of this manuscript to understand the purpose of the recorded EMG signal and its relevance for the study. Therefore, I suggest to include your response to my previous comment also in the manuscript.

In Figure 1 I would recommend consistency with the rest of the text. Now, throughout the manuscript EC and EO are named "conditions"; hence, "sessions" should be replaced with "conditions, or even removed when redundant (e.g. inside the run boxes).

Experimental design

no comment

Validity of the findings

no comment

Reviewer 2 ·

Basic reporting

The article still contains language errors, e.g.:

line 47/113: "Theses" - >These
line 179: "chosen" -> chose
line 280: didn’t -> did not

Experimental design

No comment.

Validity of the findings

The authors write in lines 186-188 that 'A surrogate permutation test (p< 0.01; 2000 permutations) from the EEGLAB toolbox was used to validate differences in terms of time-frequency and localisation of this ERSPs with a good alpha level (< 5%)'

What is a "good" alpha level? What is the different between the alpha level and the p level of the permutation test? For me, the provided information is not sufficient to reproduce the given results. Please clarify this by providing even more details.

·

Basic reporting

I am satisfied with the changes made in this aspect.

Experimental design

I am satisfied with the changes made in this aspect.

Validity of the findings

I am not entirely satisfied with the changes made regarding my last point (the BCI scenarios proposed as possible application of the results).

I would entirely remove the example of intraoperative awareness which, I repeat, has very little to share with an EC awake condition (please remove from lines 351 to 365 in the track changes version). You could eventually leave just the final sentence as introductory to the following example:
"The results obtained could be helpful for the Brain-Computer Interface (BCI) domain. Next step in this direction would be to verify that modulations observed for the EC condition are confirmed during an attempted movement - (in subjects with motor impairment)- or a motor imagery."
And then continue with the rehabilitation BCI example.

Additional comments

Overall, I am satisfied with the changes made except for the first BCI scenario proposed (intraoperative awareness) which in my opinion is out of focus and should be removed.

---

## Round 0.3 · accepted · Accept

· Academic Editor

Accept

The authors have addressed all outstanding issues.